# Effects of Cr and Mo on Mechanical Properties of Hot-Forged Medium Carbon TRIP-Aided Bainitic Ferrite Steels

**Koh-ichi Sugimoto [1],\*** , **Sho-hei Sato [2], Junya Kobayashi [3] and Ashok Kumar Srivastava [4]**

[1]  Department of Mechanical Systems Engineering, Graduate School of Science and Technology, Shinshu University, Nagano 380-8553, Japan
[2]  Department of Production Engineering, Sato Press Co., Ltd., Toyota 473-0933, Japan; s-sato@satopress.com
[3]  Department of Mechanical Engineering, Graduate School of Science and Engineering, Ibaraki University, Hitachi 316-8511, Japan; junya.kobayashi.jkoba@vc.ibaraki.ac.jp
[4]  Department of Metallurgical Engineering, School of Engineering, OP Jindal University, Raigarh 496109, India; ashok.srivastava@opju.ac.in
*  Correspondence: sugimot@shinshu-u.ac.jp; Tel.: +81-90-9667-4482

**Abstract:** In this study, the effects of Cr and Mo additions on mechanical properties of hot-forged medium carbon TRIP-aided bainitic ferrite (TBF) steel were investigated. If 0.5%Cr was added to the base steel with a chemical composition of 0.4%C, 1.5%Si, 1.5%Mn, 0.5%Al, and 0.05%Nb in mass%, the developed steel achieved the best combination of strength and total elongation. The best combination of strength and impact toughness was attained by multiple additions of 0.5%Cr and 0.2%Mo to the base steel. The excellent combination of strength and impact toughness substantially exceeded those of quenched and tempered JIS-SCM420 and 440 steels, although it was as high as those of 0.2%C TBF steels with 1.0%Cr and 0.2%Mo. The good impact toughness was mainly caused by uniform fine bainitic ferrite matrix structure and a large amount of metastable retained austenite.

**Keywords:** hot-forging; microalloying; TRIP-aided bainitic ferrite steel; retained austenite; tensile property; impact toughness

## 1. Introduction

In the past decades, first-, second-, and third-generation cold- and hot-rolled advanced high-strength steels (AHSSs) have been developed in the world [1,2]. Ferrite-martensite dual-phase steels, TRIP-aided steels with polygonal ferrite matrix structure, and complex steel are categorized as the first-generation AHSSs [1]. High-Mn twinning-induced plasticity (TWIP) steels are known as the typical second-generation AHSSs [2]. The typical third-generation AHSSs are transformation-induced plasticity (TRIP)-aided bainitic ferrite (TBF) [3], bainitic ferrite/martensite (TBM) and martensite (TM) steels [4–7], quenching and partitioning (Q&P) steels [8–10], carbide-free bainitic (CFB) steels (or nano-structured bainitic steels) [11–13], and medium Mn (M-Mn) steels [14–16]. Cold rolled AHSSs of 980−1180 MPa grade with excellent cold formability have already been applied to automotive body in white and seat frame in order to reduce the weight and enhance the crush safety [17–19]. In addition, 1180 MPa hot-rolled AHSS has been successfully applied in truck cylinders for concrete mixer [20].

Low- and medium-carbon TBF, TBM, and TM steels [4–7] are produced by a similar heat treatment process to low- and medium-carbon Q&P [8–10], CFB steels [11–13], and martensite type M-Mn steels [15], except Q&P steel subjected to two step Q&P process and dual-phase type M-Mn steels [14,16]. The heat treatment consists of austenitizing and subsequent austemper or martempering.

Recently, an interesting project for weight reduction and size-down of automotive forging parts such as Powertrain components etc., "The Lightweight Forging Initiative", was implemented in Germany [21,22]. In this project, V-bearing precipitation-hardening ferritic-pearlitic steels and bainitic steels without heat treatment after hot-forging were used on behalf of quenched and tempered martensitic steels for the weight reduction and size-down. For further weight reduction, TBF, TBM, and TM steels [23–25] are also very attractive as well as Q&P [26,27], CFB [26,28–33], and martensite type M-Mn steels [15], because their steels possess excellent mechanical properties such as tensile strength, impact toughness, fatigue strength, and delayed fracture strength.

In order to develop a new hot-forged TBF steel, Sugimoto et al. [34,35] investigated the effects of hot forging in $\gamma$ region and the subsequent austemper (FA) process on the microstructure and mechanical properties of TBF steels with chemical compositions of 0.4%C-1.5%Si-1.5%Mn and 0.4%C-1.5%Si-1.5%Mn-1.0%Al-0.05%Nb. They obtained the following interesting results:

(1)　The FA process refined the microstructure and increased the volume fraction of retained austenite with a decrease in its carbon concentration.
(2)　Good combination of yield strength and impact toughness was achieved when austemper was conducted at temperatures above $M_s$.

If the TBF steels are applied to relatively large forging parts, high hardenability may be required to obtain the mixed microstructure of bainitic ferrite and metastable retained austenite. In general, hardenability of the steel is improved by the addition of alloying elements such as Cr, Mo, Ni, Mn, B, etc. However, there is no research investigating the effects of hardenability on microstructure and mechanical properties in the hot-forged medium-carbon TBF steels.

In the present study, the effects of Cr and Mo additions on the microstructure and mechanical properties (such as tensile properties and impact toughness) of 0.4%C-1.5%Si-1.5%Mn-0.5%Al- 0.05%Nb TBF steels subjected to the FA process were experimentally investigated. The mechanical properties were related to the microstructural and retained austenite characteristics. In order to investigate the effects of carbon content, the mechanical properties were compared with those of hot-forged low-carbon TBF steels (0.2%C-1.5%Si-1.5%Mn-0.04%Al-0.05%Nb-(0–1.0)%Cr-(0–0.2)%Mo). In addition, the mechanical properties were compared with those of commercial JIS-SCM420 and 440 steels [36].

## 2. Experimental Procedure

Three 100 kg ingots of 100 mm in diameter were vacuum-melted and then hot-forged to 32 mm in diameter. Chemical compositions of the steel bars are shown in Table 1. Steel A is a base steel containing 0.40%C, 1.49%Si, 1.49%Mn, 0.49%Al, and 0.048%Nb. Al and Nb were added to the base steel to stabilize the retained austenite and refine the prior austenite grain size, respectively. Steel B was obtained by adding about 0.51% Cr to the Steel A. Steel C was produced by further adding of up to 0.20%Mo to the Steel B. For the Steels B and C, Cr and Mo were added to improve the hardenability. The martensite-start and -finish temperatures ($M_S$ and $M_f$) of the steels were measured using a dilatometer (Thermecmaster-Z, Fuji Electronic Ind. Co., Osaka, Japan). The continuous cooling transformation (CCT) diagrams of Steels A–C are shown in Figure 1. To investigate the effects of carbon content, four kinds of low-carbon 1.5%Si-1.5%Mn steels with different Cr and Mo contents (Steels D−G) were used. In addition, commercial JIS-SCM420 (0.21%C-0.26%Si-0.86%Mn-1.10%Cr-0.16%Mo) and JIS-SCM440 (0.39%C-0.19%Si-0.68%Mn- 0.95%Cr-0.17%Mo) steels [36] were also used to clear the effects of Si content.

**Table 1.** Chemical composition [mass%], martensite-start temperature $M_S$ [°C] and carbon equivalent $C_{eq}$, [mass%] of medium- and low-carbon steels used.

| Steel | C | Si | Mn | P | S | Cr | Mo | Al | Nb | N | O | $M_S$ | $C_{eq}$ |
|---|---|---|---|---|---|---|---|---|---|---|---|---|---|
| A | 0.40 | 1.49 | 1.49 | <0.005 | 0.0021 | - | - | 0.49 | 0.048 | 0.0009 | 0.0006 | 325 | 0.710 |
| B | 0.43 | 1.50 | 1.52 | <0.005 | 0.0023 | 0.51 | - | 0.49 | 0.052 | 0.0009 | 0.0005 | 318 | 0.846 |
| C | 0.42 | 1.47 | 1.51 | <0.005 | 0.0019 | 0.50 | 0.20 | 0.48 | 0.052 | 0.0010 | 0.0007 | 310 | 0.883 |
| D | 0.20 | 1.52 | 1.50 | 0.004 | 0.0021 | - | - | 0.039 | - | 0.0011 | 0.0010 | 436 | 0.513 |
| E | 0.21 | 1.49 | 1.50 | 0.004 | 0.0019 | 0.50 | - | 0.040 | 0.05 | 0.0012 | 0.0012 | 426 | 0.622 |
| F | 0.21 | 1.49 | 1.50 | 0.004 | 0.0019 | 1.00 | - | 0.040 | 0.05 | 0.0013 | 0.0012 | 416 | 0.722 |
| G | 0.18 | 1.48 | 1.49 | 0.004 | 0.0029 | 1.02 | 0.20 | 0.043 | 0.05 | 0.0010 | 0.0015 | 404 | 0.744 |

In this study, the effects of Cr and Mo were replaced by the following carbon equivalent ($C_{eq}$).

$$C_{eq} = C + Si/24 + Mn/6 + Ni/40 + Cr/5 + Mo/4 + V/24, \tag{1}$$

where *C*, *Si*, *Mn*, *Ni*, *Cr*, *Mo*, and *V* represent content in mass% of individual alloying elements.

Square bars of 20 mm thickness, 32 mm width, and 80 mm length, milled from steel bars of φ32 mm, were held at 900 °C for 1200 s and then hot-forged in one stage up to a reduction strain of 50% using a 400-ton hot-forging machine, followed by austemper at 350 °C for 1000 s (Figure 2b). The austemper temperature corresponds to an optimum temperature for impact toughness of Steels A to C [35]. Die temperature and strain rate of hot-forging were about 350 °C and about 50%/s, respectively. Other bars were subjected to the only austemper without hot-forging (Figure 2a). Hereafter, this process is called conventional austemper (CA) process. The same heat treatment as Figure 2 was conducted for Steels D−G. In this case, the austemper temperature (350 °C) is lower than $M_s$ (404−436 °C), differing from Steels A−C, because the temperature gives the best mechanical properties in Steels D−G [36]. Heat treatment of quenching in oil and then tempering at 200–600 °C for 3600 s was conducted to JIS-SCM420 and 440 steels.

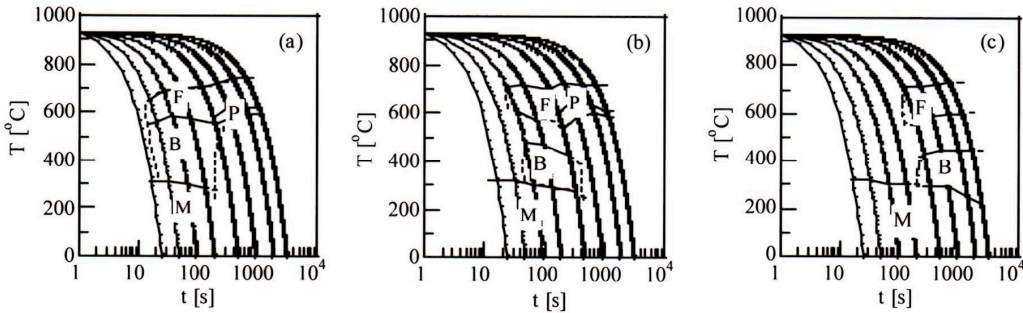

**Figure 1.** CCT diagrams of Steels (**a**) A, (**b**) B, and (**c**) C, in which "F", "P", "B", and "M" represent ferrite, pearlite, bainite, and martensite, respectively.

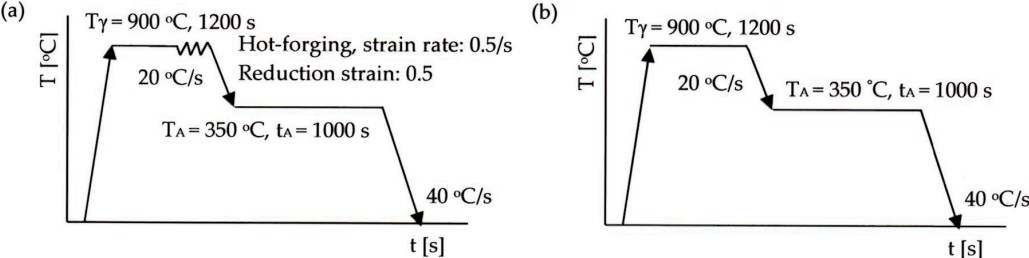

**Figure 2.** Schematic diagram of the (**a**) FA and (**b**) CA processes. *T*γ—austenitizing temperature; $T_A$—austempering temperature; $t_A$—holding time of austemper.

The microstructure of the steels was observed by field-emission scanning electron microscopy (FE-SEM; JSM-7000F, JEOL Ltd., Tokyo, Japan) using an instrument equipped with electron-backscatter diffraction (EBSD; OIM system, TexSEM Laboratories, Inc., Draper, UT, USA) system. The specimens for FE–SEM–EBSD analysis were milled with alumina powder and colloidal silica and then ion-thinned. In this case, some samples were mounted by resin and polished together to obtain the same surface condition.

The retained austenite characteristics of the steels were evaluated by X-ray diffractometry (XRD; RINT2000, Rigaku Co., Tokyo, Japan). The surfaces of the specimens were electropolished after being ground with Emery paper (#1200). The volume fraction of retained austenite phase ($f\gamma$, vol.%) was quantified from the integrated intensity of $(200)\alpha$, $(211)\alpha$, $(200)\gamma$, $(220)\gamma$, and $(311)\gamma$ peaks obtained using Mo-K$\alpha$ radiation [37]. The carbon concentration ($C_\gamma$, mass%) was estimated from the following empirical equation proposed by Dyson and Holmes [38]. Lattice constant ($a_\gamma$, $10^{-1}$ nm) was determined from the $(200)\gamma$, $(220)\gamma$, and $(311)\gamma$ peaks of Cu-K$\alpha$ radiation.

Mechanical stability of the retained austenite was calculated using the strain-induced transformation factor ($k$) defined by the following Equation [4]:

$$k = (\ln f\gamma_0 - \ln f\gamma)/\varepsilon,\tag{2}$$

where $f\gamma_0$ is an original volume fraction of retained austenite and $f\gamma$ is the volume fraction of retained austenite after being strained to plastic strain $\varepsilon$.

Tensile specimens of JIS-14B-type (22 mm length, 6 mm width, and 1.2 mm thickness) and JIS-4-type of sub-sized Charpy impact specimens with V-notch of 2 mm depth and 2.5 mm thickness were machined from 1/4 part of hot-forged plate thickness (Figure 3). Tensile tests were carried out at 25 °C using a tensile testing machine (AG-10TD, Shimadzu Co., Kyoto, Japan) under a crosshead speed of 1 mm/min. Impact tests were conducted at 25 °C using a conventional Charpy impact testing machine (CI-300, Tokyo Testing Machines Inc., Tokyo, Japan). Both tests were carried out using two specimens each.

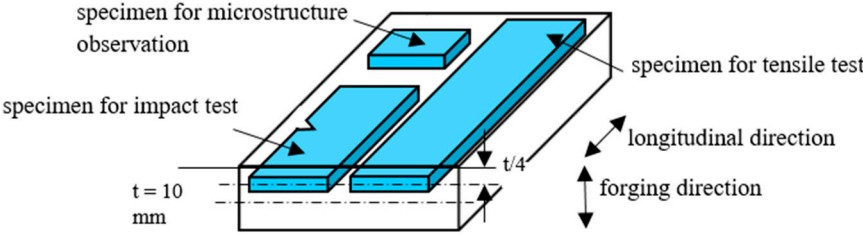

**Figure 3.** Sampling scheme of specimens from hot-forged plates. Unit of dimension is mm.

## 3. Results and Discussion

### 3.1. Microstructure and Retained Austenite Characteristics

Figure 4 shows EBSD phase maps of Steels A−C subjected to the CA and FA processes. It is found that the FA process considerably refines the sizes of prior austenitic grain, proeutectoid ferrite, bainitic ferrite, and retained austenite in Steels A−C. The higher the carbon equivalent of the steels, the more the volume fraction of acicular bainitic ferrite in the hot-forged steels. The microstructure of Steel C subjected to the FA process consists of acicular bainitic ferrite and retained austenite, with a negligible amount of proeutectoid ferrite and blacklike phase (Figure 4c,f). Refined, retained austenite phases distribute uniformly, compared with those of steel C without hot-forging. In this case, filmy- and particulate-retained austenite phases exist along the bainitic ferrite lath boundary and prior austenitic grain boundary, respectively. As the blacklike phases consist of many ultra-fine lath structures, as shown in Figure 4g–i, they are estimated to be a mixture of martensite and austenite or MA phase.

In hot-forged Steels C, the MA phase is slightly refined. The volume fraction of MA phase seems to be decreased by hot-forging.

The microstructure of Steel B subjected to the CA and FA processes seems to be complex. Considering the CCT diagram (Figure 1b), the matrix microstructure of Steel B subjected to the FA process consists of proeutectoid ferrite and acicular bainitic ferrite because of low carbon equivalent, although lath size of the acicular bainitic ferrite is larger than that of hot-forged Steel C. In the matrix structure, retained austenite and a small amount of refined MA phase are included. In Steel B without hot-forging, granular and acicular bainitic ferrites coexist in the matrix. As Steel A is characterized by the lowest hardenability, as shown in CCT diagram of Figure 1a, the matrix structures of Steel A subjected to the CA and FA processes are composed of much proeutectoid ferrite and bainitic ferrite (Figure 4a,d). Most of the bainitic ferrite seems to be granular. The retained austenite size is the largest of Steels A to C. The FA process increases MA phase fraction.

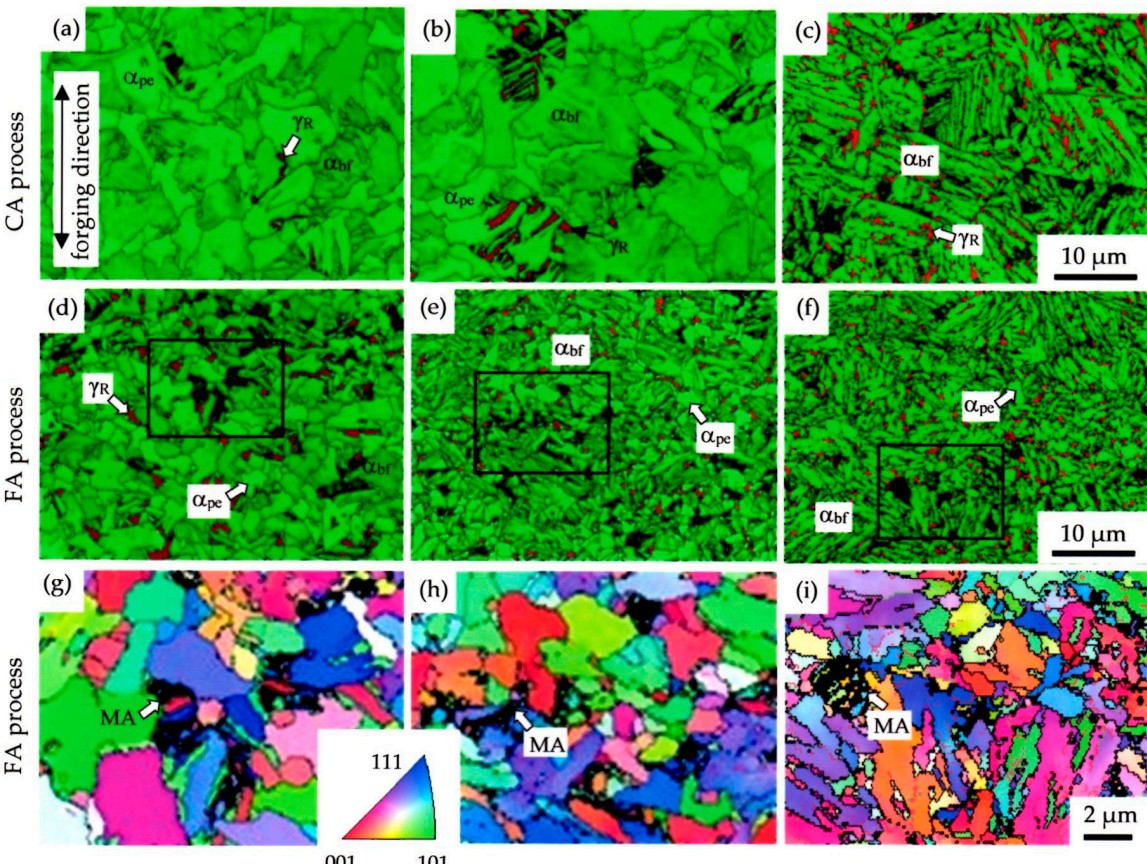

**Figure 4.** EBSP phase maps of Steels A (**a,d,g**), B (**b,e,h**) and C (**c,f,i**) subjected to CA and FA process. (**g–i**) are high magnification orientation maps of squares in (**d–f**), respectively. "$\alpha_{pe}$", "$\alpha_{bf}$", "$\gamma_R$" and "MA" represent pro-eutectoid ferrite, acicular bainitic ferrite, retained austenite (red) and martensite-austenite phase, respectively.

In general, image quality (IQ) index of the phase map by EBSD analysis can be related to dislocation density of individual phases [39]. Figure 5 shows IQ indices of FCC and BCC phases of Steels A and C without and with hot-forging. Peak IQ indices of both phases in Steel A subjected to the FA process are nearly equal to those after the CA process. This indicates that recrystallization after hot-forging finishes perfectly in both phases of Steel A. On the other hand, the peak IQ indices of both phases in Steel C shift to low index by FA process, different from Steel A. This indicates that bainitic ferrite and retained austenite phases in hot-forged Steel C inherit, to some extent, high dislocation density resulting from

hot-forging via interaction between ultra-fine carbonitrides of Cr, Mo, and/or Nb. This should be further investigated using TEM analysis.

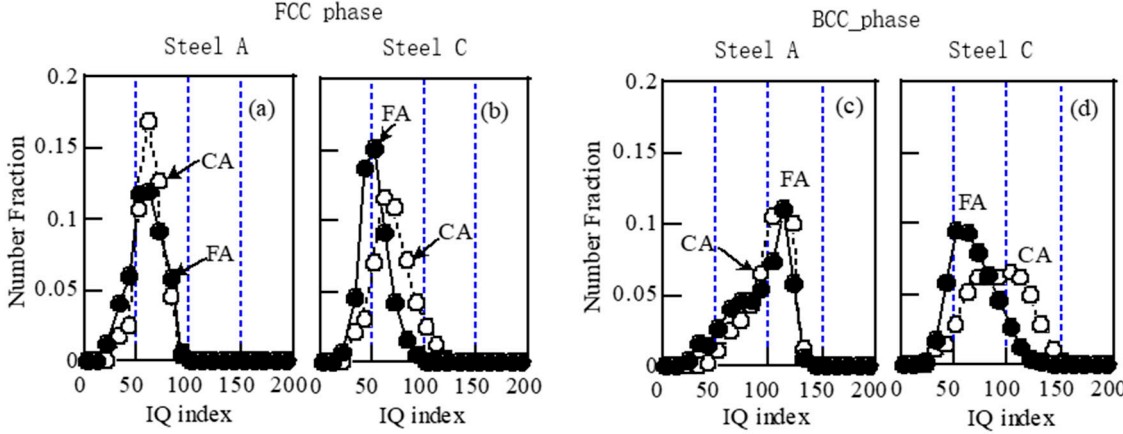

**Figure 5.** Image quality (IQ) distribution of (**a**,**b**) FCC and (**c**,**d**) BCC phase in Steels A and C subjected to CA and FA processes.

Figure 6a–d show an initial volume fraction ($f\gamma_0$), carbon concentration ($C\gamma_0$), and total carbon concentration ($f\gamma_0 \times C\gamma_0$) of retained austenite and a ratio of $f\gamma_0 \times C\gamma_0$ to added C content ($f\gamma_0 \times C\gamma_0$/C), as a function of carbon equivalent in Steels A to C. The volume fraction and carbon concentration increase with increasing carbon equivalent in hot-forged Steels A to C, as well as the total carbon concentration. The FA process tends to decrease the carbon concentration and total carbon concentration of retained austenite in comparison to CA process, although the FA process slightly increases the volume fraction of retained austenite except for Steel B. The $f\gamma_0 \times C\gamma_0$/C, which means a percentage solute carbon dissolved in retained austenite, increases with increasing carbon equivalents in Steels A to C subjected to FA process, although the FA process decreases the values.

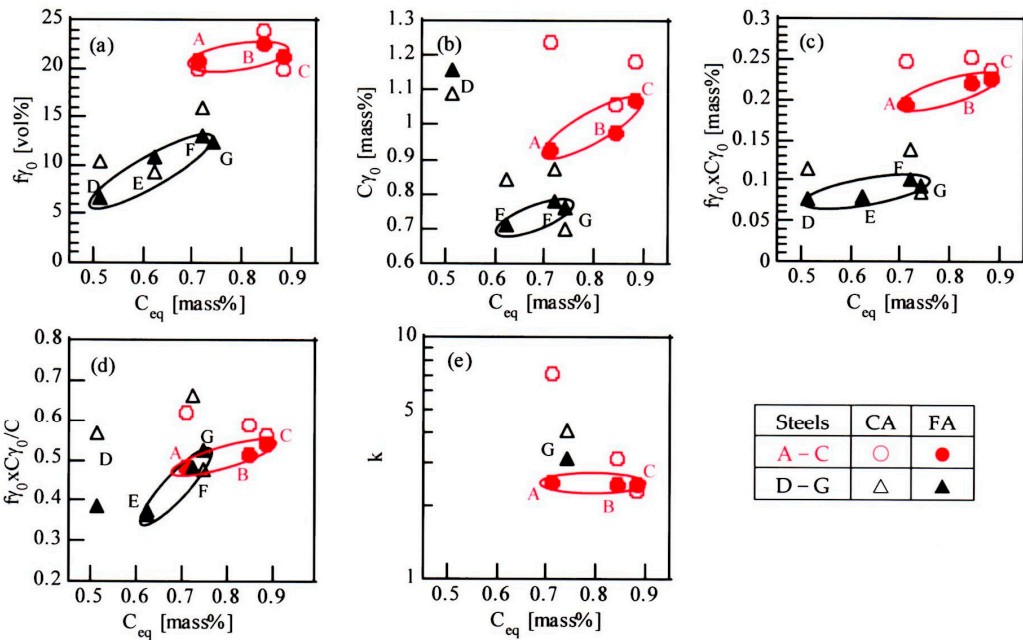

**Figure 6.** Carbon equivalent ($C_{eq}$) dependences of (**a**) initial volume fraction ($f\gamma_0$), (**b**) initial carbon concentration ($C\gamma_0$), (**c**) initial total carbon concentration ($f\gamma_0 \times C\gamma_0$) of retained austenite, (**d**) a ratio of $f\gamma_0 \times C\gamma_0$ to added C content ($f\gamma_0 \times C\gamma_0$/C), and (**e**) apparent *k*-value in Steels A−C and D−G subjected to the CA and FA processes.

Figure 6e shows apparent *k*-values of Steels A to C. The FA process tends to make the *k*-values smaller, particularly in Steel A, despite the low carbon concentration of retained austenite. This result indicates that the mechanical stability of retained austenite is principally decided by retained austenite size, compared with the carbon concentration, because the refined retained austenite suppresses the strain-induced martensite transformation [35,36]. It is found that Steels A to C subjected to the FA process apparently exhibit same *k*-values (or same mechanical stability of retained austenite). As the *k*-value of Steel C is evaluated at higher flow stress, the true mechanical stability is the highest of Steels A to C.

When the retained austenite characteristics of Steels A to C were compared with those of Steels D to G, Steels D to G were found to have lower carbon content, volume fraction, carbon concentration, and mechanical stability of the retained austenite than Steels A to C. It is noteworthy that carbon equivalent dependences of the retained austenite characteristics in Steels D to G are nearly the same as those of Steels A to C.

*3.2. Tensile Properties*

Figure 7 shows flow curves of Steels A, B, and C subjected to the CA and FA processes. Figure 8a–d show the tensile properties as a function of carbon equivalent. All steels exhibit continuous yielding and subsequent large strain hardening, especially in Steels A and C subjected to FA process. Yield stress or 0.2% offset proof stress (YS), tensile strength (TS), and yield ratio (YR = YS/TS) increase with increasing carbon equivalent in the steels subjected to the CA and FA processes. The FA process increases these strengths and decreases the yield ratio due to the increased strain-hardening rate.

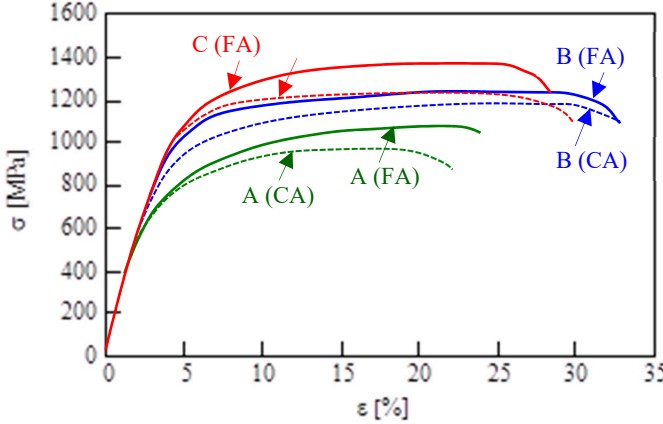

**Figure 7.** Nominal stress–strain curves of Steels A (green curves), B (blue curves) and C (red curves) subjected to CA and FA processes.

Carbon equivalent dependence of total elongation (TEl) is indistinct in Steels A to C subjected to the CA and FA processes. However, if Steel A is excluded, the total elongation decreases with increasing carbon equivalent, contrary to yield stress and tensile strength. Regarding total elongation, the FA process decreases the total elongation in Steel C depending on the trade-off relationship with tensile strength, although a difference in total elongations between the CA and FA processes is relatively small. As shown in Figure 8a,b, the largest combinations of yield stress and total elongation (YS × TEl), and tensile strength and total elongation (TS × TEl), are achieved in hot-forged Steel B. When these combinations were compared to those of hot-forged Steels D−G and JIS-SCM420 and 440 steels, hot-forged Steel B exhibits higher combinations than those of comparable steels, as well as hot-forged Steel C. In general, MA phase deteriorates the total elongation as an initiation site of void or crack. However, it is well known that the strain-induced transformation of retained austenite suppresses the void and/or crack initiation through the plastic relaxation effect in TBF steel [23]. Thus, it is considered that the MA phases do not decrease the total elongation.

According to Sugimoto et al. [35,36], tensile properties of hot-forged low- and medium-carbon TBF steels are influenced by the size and type of matrix structure, prior austenitic grain size, second phase properties, and retained austenite characteristics. In the present study, volume fraction of acicular bainitic ferrite increased with increasing carbon equivalent in Steels A to C (Figure 4). As the acicular bainitic ferrite is stronger than granular bainitic ferrite and proeutectoid ferrite, positive equivalent carbon dependences of yield stress and tensile strength (Figure 8a,b) may be mainly associated with the increased volume fraction of acicular bainitic ferrite. The increases in yield stress and tensile strength by hot-forging is supposed to be mainly caused by refined microstructure. High dislocation density of acicular bainitic ferrite and retained austenite (Figure 5) contributes to the highest yield stress and tensile strength obtained in Steel C.

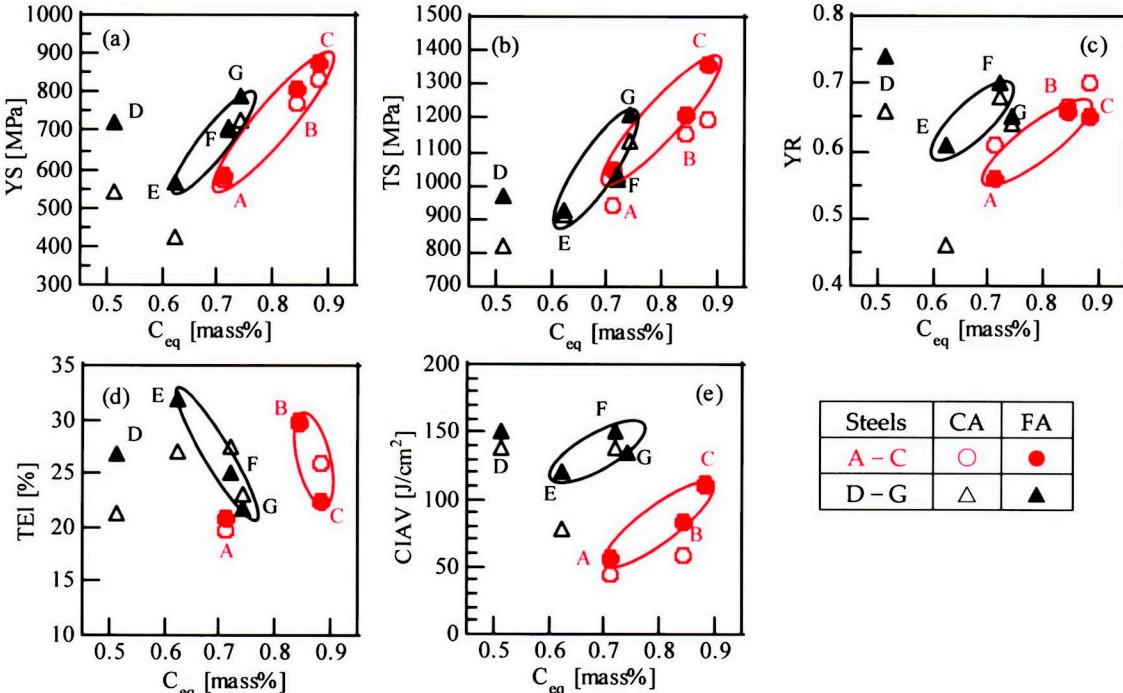

**Figure 8.** Carbon equivalent ($C_{eq}$) dependences of (**a**) yield stress (YS), (**b**) tensile strength (TS), (**c**) yield ratio (YR = YS/TS), (**d**) total elongation (TEl), and (**e**) Charpy impact absorbed value (CIAV) in Steels A−C and D−G subjected to the CA and FA processes.

The maximum total elongation and combinations YS × TEl and TS × TEl were achieved in Steel B subjected to the FA process (Figures 8d and 9a,b). Steel B possessed a dual-phase structure of proeutectoid ferrite and acicular bainitic ferrite. The dual-phase structure develops a compressive long-range internal stress in soft proeutectoid ferrite, differing from a single phase of acicular bainitic ferrite in Steel C. Resultantly, the internal stress may keep high strain-hardening rate in a large strain range and suppresses the diffuse necking, as well as an increase in flow stress [40]. Likewise, a large amount of metastable retained austenite enhances the strain-hardening rate via the TRIP effect and as a hard phase, especially in a large strain range. Therefore, excellent ductility of Steel B is considered to be brought by the dual-phase structure and a large amount of metastable retained austenite.

In Figure 9a,b, the combinations YS × TEl and TS × TEl of Steel B subjected to FA process was higher than those of Steels D−G subjected to FA process, as well as SCM420 and 440 steels. This may be associated with high retained austenite fraction and mechanical stability (Figure 6a,e). It is noteworthy that the combinations YS × TEl and TS × TEl of Steels D−G are of the same degree regardless of Cr and Mo contents.

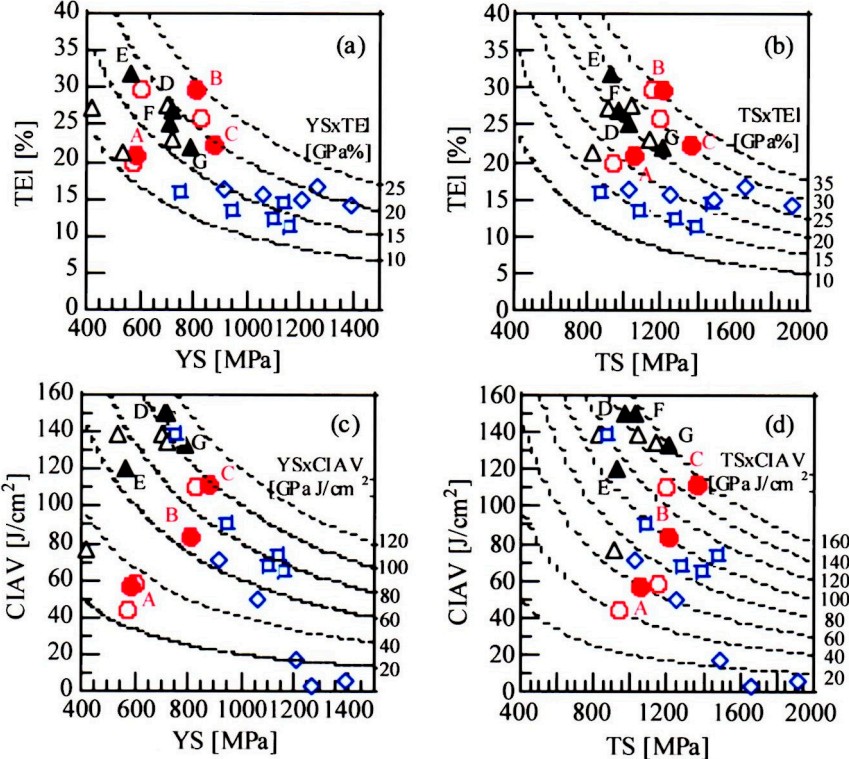

**Figure 9.** Combinations of (**a**,**b**) total elongation (TEl) with yield stress (YS) and with tensile strength (TS), respectively, and (**c**,**d**) V-notch Charpy impact absorbed value (CIAV) with YS and with TS, respectively, in Steels A−C (○●) and Steels D−G (△▲) subjected to the CA (open marks) and FA (solid marks) processes and quenched and tempered SCM420 (□) and 440 steels (◇).

### 3.3. Impact Toughness

Figure 8e shows Charpy impact absorbed value (CIAV) as a function of carbon equivalent in Steels A to C subjected to the CA and FA processes. The CIAV increases with increasing carbon equivalent in hot-forged Steels A to C. The FA process increases the CIAVs. The CIAVs are much lower than those of Steels D–G. Figure 9c,d show the combinations of YS and CIAV (YS × CIAV) and TS and CIAV (TS × CIAV) in Steels A−C subjected to the CA and FA processes. Maximum combinations are achieved in Steel C. The maximum combinations are the same extent as those of Steels F and G and are much higher than those of SCM420 and 440 steels. It is noteworthy that the single addition of Cr is very effective to enhance the combinations in low-carbon TBF steels (Steels D to G), differing from medium-carbon steels (Steels A to C).

Figure 10 shows SEM images of ductile fracture region and optical micrographs of fracture surface appearance in Steels A−C subjected to the CA and FA processes. High percent ductile fracture (PDF) can be related with high CIAV of these steels. Namely, the PDF increases with increasing CIAV, except Steel B subjected to the CA process. The FA process increases the PDF. It is found in the figure that the FA process makes the void size fine and void distribution uniform. As the PDF values of Steels A to C subjected to the FA process are 100%, ductile–brittle transition temperatures of all the steels are indicated to be lower than room temperature, as well as Steel C subjected to the CA process.

As with the ductility, impact toughness of low- and medium-carbon TBF steels is affected by size and type of matrix structure, prior austenitic grain size, second-phase properties, and retained austenite characteristics, which control the initiation and growth behavior of void and/or crack [23–25,41]. In this case, metastable retained austenite plastically relaxes the localized stress concentration through expansion strain on the strain-induced martensite transformation. Simultaneously, the strain-induced martensite makes a defense against crack initiation and propagation and brings in high-impact energy at

an initial stage (strain-hardening stage) [41]. On the other hand, carbide-free acicular bainitic ferrite lath structure, a large amount of metastable retained austenite, and refined prior austenitic grain boundary suppress the initiation and growth of void or crack by lowering the localized stress concentration. This resultantly brings in high-impact energy at a final stage from top stress to fracture. In this case, carbon-enriched retained austenite in MA phases may also contribute to the impact energy through the plastic relaxation [23]. These microstructural features also lower the ductile–brittle transition temperature. From the above theory and the results of Figures 4 and 6e, high CIAV and combinations YS × CIAV and TS × CIAV of hot-forged Steel C may be mainly caused by refined uniform acicular bainitic ferrite matrix and a large amount of metastable retained austenite. Low CIAVs of Steels A and B may be associated with the existence of much proeutectoid ferrite, because the volume fraction and mechanical stability of the retained austenite of Steels A to C are nearly the same. The lowest CIAV of hot-forged Steel A is due to relatively coarse microstructure, because it plays as a stress concentration site.

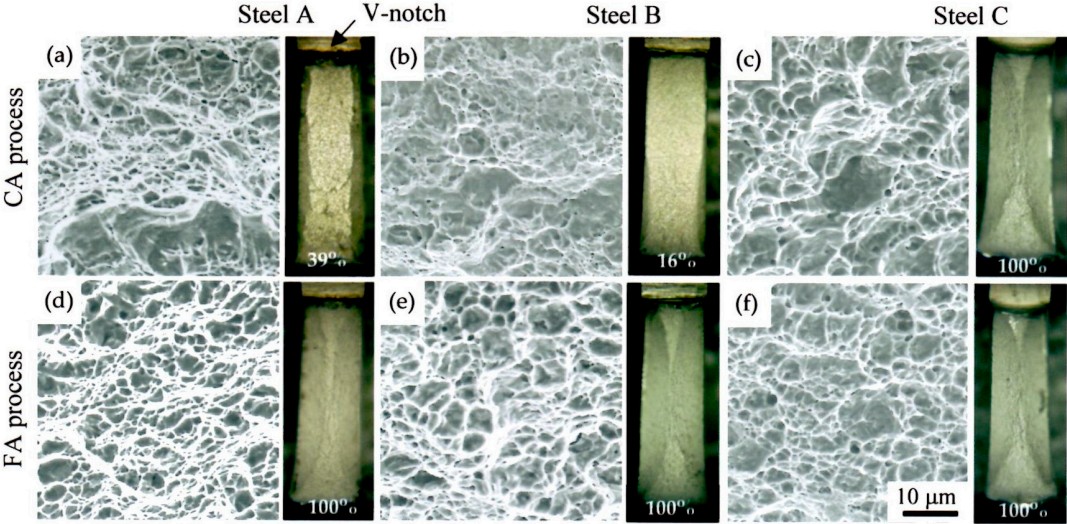

**Figure 10.** Typical SEM images of ductile fracture region at notch tip and optical micrographs of fracture surface appearance in Steels A (**a**,**d**), B (**b**,**e**), and C (**c**,**f**) subjected to the CA and FA processes. The numerals in the figure represent percent ductile fracture.

It is well known that impact toughness of medium-carbon steel considerably decreases compared to that of low-carbon steel [42]. In spite of medium-carbon content of 0.4%, Steel C possessed a relatively high CIAV (Figure 8e). Resultantly, Steel C achieved as high combinations YS × CIAV and TS × CIAV as Steels F and G with 0.2%C (Figure 9c,d). This may be caused by higher volume fraction and mechanical stability of retained austenite than those of Steels F and G (Figure 6), as well as carbide-free bainitic ferrite matrix structure. It is well known that Hy-tuf steel (0.24%C-1.42%Si-1.35%Mn-0.31%Cr-1.71%Ni-0.40%Mo-0.16%Cu) with 15–18 vol% retained austenite—which is usually applied to the aircraft parts, mining equipment, and other high-performance products—possesses high impact toughness [43]. However, the combinations YS × CIAV and TS × CIAV of the Hy-tuf steel are lower than those of hot-forged Steel C. This may be owing to the decreased carbon concentration (or low mechanical stability) of retained austenite.

## 4. Summary

The effects of Cr and Mo on the microstructure and mechanical properties of hot-forged 0.4%C-1.5%Si-1.5%Mn TBF steel were investigated. The main results are summarized as follows:

(1)　Multiple additions of Cr and Mo developed a mixed structure of uniform bainitic ferrite matrix and a large amount of retained austenite in the TBF steel, with a negligible amount of proeutectoid

ferrite and MA phase. Hot-forging refined the mixed structure and increased the volume fraction and mechanical stability of retained austenite. The single addition of Cr reduced the proeutectoid ferrite fraction in the mixed structure, and it achieved the maximum retained austenite fraction.

(2) Multiple additions of Cr and Mo increased yield stress and tensile strength in the TBF steels. Hot-forging furthered the yield stress and tensile strength. The single addition of Cr achieved the maximum total elongation and combinations YS × TEl and TS × TEl because of a dual-phase structure of proeutectoid ferrite and acicular bainitic ferrite and high retained austenite fraction. In this case, multiple additions of Cr and Mo achieved slightly lower combinations than the single addition of Cr.

(3) Multiple addition of Cr and Mo brought on the maximum impact toughness and combinations YS × CIAV and TS × CIAV in hot-forged TBF steels. The combinations were the same as those of low-carbon TBF steels with chemical composition of 0.2%C-1.5%Si-1.5%Mn- 0.5%Cr-(0−0.2)%Mo. The excellent impact toughness was principally caused by uniform fine acicular bainitic ferrite matrix structure and a large amount of metastable retained austenite. It was considered that the mechanically stable retained austenite is especially effective to improve the impact toughness due to (i) the plastic relaxation of localized stress concentration and (ii) the increased martensite fraction via the strain-induced martensite transformation.

**Author Contributions:** Microstructure examination, tensile tests and impact tests were mainly performed by S.S. and J.K., the first draft of the paper was written by K.S. and A.K.S. All authors contributed to editing the paper, with final edits by K.S. and A.K.S.

**Funding:** This research was supported by a Grant-in-Aid for Scientific Research (B), The Ministry of Education, Science, Sports and Culture, Japan (No. 201325289262).

**Acknowledgments:** We thank Goro Arai from Nomura Unison Co., Ltd. for hot-forging tests.

**Conflicts of Interest:** The authors declare no conflict of interest.

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
