# Peer review of "Effects of Cr and Mo on Mechanical Properties of Hot-Forged Medium Carbon TRIP-Aided Bainitic Ferrite Steels"

_metals, doi:10.3390/met9101066_

Round 1

Reviewer 1 Report

In general, the mauscript is well written and the study was well designed. The overview of the literature is nicely presented. Methods are clearly explained. The results are very interesting.

I see only one minor issue that should be clarified before publishing.

Concerning Figure 4, row 176 - "In the matrix structure, retained austenite and a small amount of undetectable phase are included". The authors  should use TEM analysis or SEM (mapping) analysis to identify this  undetectable phase.

Figure 5, row 206-209 - "On the other hand, the peak IQ indices of both
phases in Steel C shift to low index by FA process, different from Steel A. This indicates that bainitic ferrite and retained austenite phases in hot forged Steel C inherit to some extent high dislocation density resulting from hot forging". I would suggest TEM analysis to be performed because this would give more information about interaction between dislocation and precipitation, especially precipitate of Cr, Mo, Nb (i.e small nitrides, about 10nm in size). The results of TEM analysis could improve the conclusion of this paper. The authors should report if TEM analysis was done, and if so, the results should be also reported in this paper. If not, I suggest to do this in further investigations.

Author Response

Very thanks for your valuable comments. Please review our revised manuscript again.

Reviewer 2 Report

Dear Authors,

Thank you for sharing such a nice work on the field of TBF.

Overview: the work explores the influence of Cr and Mo additions on a given TBF steel, comparing the effect of forging+austempering against conventional austempering. A selection of steels with lower C and Al is used for comparison together with two standard Q+T steels.

General comments:

The work is very well presented, even thought nine distinct chemical compositions and two processes (CA and FA) are explored,  and the results help understanding the influence of the Cr and Mo additions. Nevertheless, clarifying a bit better the reasons behind the selection of Steels D, E, F, G, SCM420 and SCM440 for comparison and their significance in the frame of the work would greatly help the reader to go through the paper. This information could fit well in the "Introduction" or the "Experimental Procedure".

Specific comments:

Line 70. Initial hot-rolling reduction of the raw material would be interesting to be known for comparison with usual reductions for Ø32 forging stock.

Line 72. Where it says "miner" should say "minor".

Table 1. Including the nominal composition for a SMC420 and SMC440 would help, as the JIS standard may be not known by the reader.

Line 90. Different raw material was used for 20x32mm squares or were they milled from Ø32 the same bars that were used for forging? Please clarify if possible.

Line 96. Shouldn't steels D-G have been treated to their most convenient austempering temperature for proper comparison? Why was it decided to use the same conditions as for A-C?

Line 99. Tempering temperature information covers a too wide range. Temper embrittlement could be happening and obscuring the conclusions on TEl and CIAV. Would it be possible to be more specific?

Figure 4. Some of the boxes are have moved and are difficult to read in the picture. Please check if fixing is possible.

Figure 8.  Would it be possible to indicate how many specimens of each condition were used and include an uncertainty bar? Results seem to be very close to each other for CA and FA processes in A-C.

Line 373. At first sight it seems that some of the CA pictures are against the statements on void size and distribution. It might be that the selected pictures are not the most favourable to support the actual observations. Would it be possible to select fields which are more representative of the results?

Line 421. Where it says "forger" should say "forged".

Thanks for sharing your work. I really look forward to receiving the answers to my comments.

Regards.

Author Response

(The authors gave the same response as above.)

Reviewer 3 Report

The materials were treated with two different heat treatment regimes: conventional austempering without hot forging and hot forging with subsequent austempering. The manuscript contains a lot of results such a metallographic analysis, mechanical properties (UTS and impact toughness), and so on.

For the authors, I have the following recommendations and comments:

Improve the abstract. In the first sentence is written that the effect of Cr and Mo was observed. But in the second sentence, only the impact of Mo is mentioned. The influence of Cr is mentioned once again on line 18. And so on…. On line 29 is written that the typical AHSS steels are also TRIP steels, martensite steels, Q-P steels and so on. But these steels are more from the first or second generation. The third generation is more about the medium Mn steels, which use the combinations of the steels from first and second generations. The first sentence in the second paragraph in the introduction should be improved. Describe more in detail the commercially produced steels used for the comparison. On line 70 it is written that hot-rolled bars with 32 mm in diameter were used for the experiment. But on line 90 it is written about square bars. Not much information is given for other steels. Is that right? The description of heat treatment parameters is confusing. I recommend swapping figures 2 a and 2b to match the order in the text. The arrows in Figure 4 are poorly visible. There is an empty box in figure 2D. I think it would be better to compare only the effect of Mo and Cr in the first group in the experimental program. The results are mostly given only for these steels. Not much information is given for other steels (D-G). There no information about structure development after heat treatment. The results from retained austenite characterization could be found for these steel in figure 6 but without description in the text.  Figure 9 is confusing. The text uses a lot of abbreviations.

Author Response

(The authors gave the same response as above.)

Round 2

Reviewer 2 Report

Dear authors, thanks for addressing my comments. 

Regards